# Public attitudes towards neurotechnology: Findings from two experiments concerning Brain Stimulation Devices (BSDs) and Brain-Computer Interfaces (BCIs)

**Sebastian Sattler** [1,2,3]*, **Dana Pietralla** [2,4]

**1** Faculty of Sociology, Bielefeld University, Bielefeld, Germany, **2** Institute of Sociology and Social Psychology, University of Cologne, Cologne, Germany, **3** Pragmatic Health Ethics Research Unit, Institut de Recherches Cliniques de Montréal, Montréal, Canada, **4** Department of Psychology, University of Cologne, Cologne, Germany

* sebastian.sattler@uni-bielefeld.de

**Data Availability Statement:** The data that support the findings of this study are openly available in "PUB-Publikationen an der Universität Bielefeld"

## Abstract

This study contributes to the emerging literature on public perceptions of neurotechnological devices (NTDs) in their medical and non-medical applications, depending on their invasiveness, framing effects, and interindividual differences related to personal needs and values. We conducted two web-based between-subject experiments (2×2×2) using a representative, nation-wide sample of the adult population in Germany. Using vignettes describing how two NTDs, brain stimulation devices (BSDs; $N_{Experiment\ 1}$ = 1,090) and brain-computer interfaces (BCIs; $N_{Experiment\ 2}$ = 1,089), function, we randomly varied the purpose (treatment vs. enhancement) and invasiveness (noninvasive vs. invasive) of the NTD, and assessed framing effects (variable order of assessing moral acceptability first vs. willingness to use first). We found a moderate moral acceptance and willingness to use BSDs and BCIs. Respondents preferred treatment over enhancement purposes and noninvasive over invasive devices. We also found a framing effect and explored the role of personal characteristics as indicators of personal needs and values (e.g., stress, religiosity, and gender). Our results suggest that the future demand for BSDs or BCIs may depend on the purpose, invasiveness, and personal needs and values. These insights can inform technology developers about the public's needs and concerns, and enrich legal and ethical debates.

## Introduction

Advancements in neurotechnological devices (NTDs) have grown exponentially in the pioneering spirit of research and entrepreneurship [1–3]. One example of NTDs are brain stimulation devices (BSDs), which use externally applied (e.g., on a headband), non-invasive electrodes, as in the case of transcranial direct current stimulation [4], or invasive implants, as in deep brain stimulation [5]. These have been found to improve cognitive functions in memory and decision-making tasks by stimulating the brain through electrical impulses [6, 7],

[Reference 51: https://doi.org/10.4119/unibi/2966231].

**Funding:** This work was supported by the German Research Foundation [grant numbers: SA 2992/2-1 to Mr. Sebastian Sattler and ME 2082/8-1 to Mr. Guido Mehlkop; URL: https://www.dfg.de].

**Competing interests:** The authors have declared that no competing interests exist.

helping treat illnesses and ailments, such as chronic pain, epilepsy, Parkinson's disease, and depression [3, 8].

Brain-computer interfaces (BCIs) are another promising form of NTDs. They use wearable or implanted electrodes to connect the central nervous system with computers that convert brain activity into artificial output signals [9]. These signals allow a real-time control of external devices, such as wheelchairs, body prostheses, or computer programs, that support functioning in disabled people [10]. This makes BCIs beneficial for clinical applications in repairing or mitigating motor, cognitive, emotional, and perceptual impairments in, for example, the context of a stroke [11, 12].

BSDs and BCIs can be also used in the absence of medical conditions to alter or enhance key human features beyond their "normal" functioning by augmenting learning rates and other cognitive tasks [5, 13]. While it is debated what actually constitutes enhancement and where to draw the line in treatment, for example, as in restorative applications [14], the respective devices are becoming increasingly available on the consumer market [1]. They are also being developed more frequently in do-it-yourself communities for work, gaming, sports, or educational purposes [4]. Both medical and enhancement uses have sparked debate among scientists and ethicists about the implied ethical, legal, and social issues [10, 15–17]. Specifically, these issues include problems of brain hacking, informed consent, autonomy, personhood, stigma, or possible side effects, which warrant particular attention if used for enhancement becomes more prevalent [13, 18].

An assessment of NTDs that is regulated, informed, and democratic can advance the neuroethical debate and offer insights on how NTDs should be appropriately used and managed [19, 20]. Thus far, studies have begun exploring NTD usability and other related topics by surveying researchers' opinions, public concerns [12, 21], and patients' needs [22–24].

However, public attitudes on NTDs are still underexplored. Assessing the moral acceptability of using NTDs can create insights into societal norms and the laypersons' understanding of them as personally or socially good or bad [25], reflecting their hopes and fears. Moral evaluations and measures capturing the willingness to engage in certain behaviors are also strong predictors of health- and non-health-related behavior, since they capture motivational factors for acting in a certain way [26–29]. This makes them useful in assessing new phenomena and trends, as in the context of emerging NTDs [30, 31]. How the public evaluates different uses of NTDs (i.e., treatment vs. enhancement) needs further understanding since it might depend on the type of NTD, how NTDs operate, and which population is asked [8, 15, 17]. Therefore, this study investigates whether the moral acceptability and the willingness to use BSDs and BCIs depends on the purpose of use, their invasiveness, and framing effects (i.e., the assessment order of moral acceptability and use willingness). To better understand diverging public attitudes, this study also examines whether moral acceptability and willingness to use can be attributed to interindividual differences related to personal needs and values, as well as whether such differences modulate the role of the use purpose (in an exploratory analysis, we tested for a potential three-way interaction between purpose, invasiveness, and framing effects, but did not find such interaction effects; therefore, we do not discuss this further).

## Furthering the understanding of diverging public attitudes towards NTDs

**The role of the use purpose.** Until recently, prescription drugs as a form of neurotechnology have been the focus of research on perceptions regarding treatment and enhancement uses [32, 33]. Now, it is increasingly of interest whether and how this applies to a wider range of NTDs, such as BSDs and BCIs, because existing research has suggested that such differences may exist. Research on patients who have impaired mobility (e.g., Amyotrophic lateral

sclerosis) has found enthusiasm for NTDs among participants, who reported high hopes for increasing their mobility, interactions, and communication [10, 12].

While people more-readily accepted risks and ethical concerns when using NTDs for treatment [34], using them for enhancement seems to be perceived more negatively [8, 25]. Despite its negative reception, the enormous increase in companies producing consumer NTDs [35] and their rapid increase in online sales may indicate a rising willingness, need, and curiosity to own and use them [36]. In the current study, we want to investigate the use of NTDs for treatment and enhancement purposes. The latter might be associated with more concerns about necessity and the potential negative consequences of using NTDs to create "super" humans and more intense social competition [17, 25, 33, 37]. Despite the growing interest in using NTDs for enhancement, we expect a higher acceptability and willingness to use them for treatment [9, 25, 32, 33].

**The role of invasiveness.** According to Kögel et al. [10], investigations of attitudes towards NTDs must consider their method of operation. Invasive and noninvasive methods come with different risks, such as ethical concerns and varying levels of reliability [5]. These concerns differ among groups: While research with patients has found a relatively high willingness to use invasive BSDs [34], research on the public indicates more worries about invasive enhancements, and a greater magnitude of worries for its use as a treatment [25, 38]. The public's stronger rejection of invasive methods may reflect their perception that such methods are dangerous or unnatural [8, 25, 39, 40]. Therefore, we expect a higher moral acceptability and willingness to use noninvasive than invasive NTDs. Moreover, the potential risks and doubts related to invasive NTDs may reduce the acceptability and willingness to use them for enhancement rather than treatment purposes, for which risks and problems might be more tolerated. Therefore, we expect invasiveness to amplify the negative effects of treatment purposes.

**The role of framing effects.** Thinking about the morality of a behavior, e.g., using NTDs, may activate concerns that reduce the willingness to use, while moral reasoning should be more stable. Seminal research suggests that if moral acceptability was assessed prior to use willingness (versus the opposite order), respondents showed a lower willingness to use NTDs [41]. Such a framing effect has been shown for enhancement uses (especially for noninvasive BSD), but it is unknown whether informing individuals about moral concerns of NTDs may reduce their acceptance for treatment [41]. Our study aims at replicating this first evidence and testing whether such effects also exist for treatment purposes. We expect that such a framing effect could be more pronounced for enhancement purposes, since activating moral concerns against enhancement uses might be more impactful than for treatment uses, which might be seen as less abdicable.

**Interindividual differences in attitudes towards NTDs.** First studies have also tried to understand the characteristics of respondents supporting or opposing NTDs. This research has yielded inconclusive results and has hardly investigated whether attributes of the NTDs are judged differently by different people [10, 38]. However, respondents' individual perspectives and circumstances may affect their attitudes towards NTDs, as indicated by the research on prescription drugs used for enhancement, and less frequently for NTDs. Higher moral acceptance and use willingness can be expected among individuals who would have a greater need for NTDs, for example, to deal with stress and its negative consequences [29, 42–44]; to compensate for potential deficits, such as low levels of cognitive function [45]; and to cope with age-related cognitive and physical decline (such as vision or mobility) [46, 47].

Values and ideologies may also shape the moral evaluation and the willingness to use NTDs. For example, some might more strongly reject these technologies on the basis that they are "unnatural" and they transgress human boundaries. Others might have a lower affinity for

technology, which is more likely in religious [8, 17, 48], older [17, 25, 32, 41], less educated [49], or female [17, 41] populations. Therefore, we examine the role of chronic stress, cognitive functioning, religiosity, age, education, and gender as indicators of needing to use NTDs and individuals' values and ideologies towards such use. To gain more detailed insights about their role, we also investigate if individuals with certain respondent characteristics evaluate enhancement and treatment differently, since a need for using NTDs, or the values and ideologies towards them, may modulate the moral evaluation and use willingness of both uses differently (while one might argue that individuals with different characteristics also vary in their evaluation of whether NT is invasive vs. noninvasive, our exploratory analysis did not reveal any statistically significant interaction effects between the invasiveness factor and each investigated characteristic; thus, we do not further elaborate on this in the manuscript).

## The current study

Overall, previous research has offered a first insight into public attitudes towards NTDs. However, this research had typically used qualitative designs, was descriptive or correlational, mostly only examined users of NTDs, or used measures like interest in NTDs. The existing quantitative studies often used small or nonrepresentative and self-selected samples, or used within-person experiments that can cause learning and contrast effects [50]. While discussion about cognitive enhancement drug use has increased [4], especially regarding student populations, more data on the public's attitudes regarding enhancement with NTDs are needed. Moreover, because the development of NTDs occurs rapidly and the knowledge about them changes, studies dating back even a few years deserve an updated view. Therefore, our study aims at increasing the understanding of the moral acceptability and willingness to use BSDs and BCIs, since both are emerging NTDs and they partially overlap (e.g., using technology that interacts with the brain), but also have distinct characteristics and uses (e.g., BSDs stimulate the brain, while BCIs can also control external devices). By investigating two NTDs, this study facilitates a better understanding of how public attitudes in Germany relate to both NTDs and the investigated factors. Using randomly selected samples of the general adult population in Germany, we conducted two between-subject experiments. This allows for a more generalizable, causally-oriented interpretation of the findings. We examined whether the moral acceptability and use willingness differ between using NTDs for treatment or enhancement purposes and noninvasive or invasive applications, as well as the possibility of a question-order-based framing effect. Moreover, we explored the role of various personal characteristics and whether they condition the effect of the purpose of use.

## Methods

### Participants

We conducted two web-based survey experiments with participants from a nationwide sample of German-speaking adult residents in Germany. The sample was recruited offline via the forsa.omninet panel, which offers a representative sampling of the population in Germany through a multi-stage random process based on an ADM telephone master sample (i.e., ADM stands for *Arbeitskreis Deutscher Markt- und Sozialforschungsinstitute e.V.*, the Association of German Market and Social Research Institutes). Thereby, every household in Germany has the same statistical chance of participation, infrequent internet users are captured, and self-selection into the panel is avoided. Our experiments were part of a larger study (*ENHANCE*) aiming for a greater heterogeneity of participants compared to student or crowdsourced samples. In the survey, individuals were randomly assigned to one of the experiments. Participation was voluntary and those who completed it received bonus points as incentives, which could be

**Table 1. Pairwise correlations, reliability scores (of scales), and descriptive statistics of the independent variables from Experiment 1 and Experiment 2.**

| | Pairwise correlations and *Cronbach's α* in the diagonal | | | | | | Descriptive statistics | |
| --- | --- | --- | --- | --- | --- | --- | --- | --- |
| | 1) | 2) | 3) | 4) | 5) | 6) | Mean/Proportion (SD) | Min-Max |
| **Experiment 1 (*N* = 1,090)** | | | | | | | | |
| 1) Stress | *0.79* | | | | | | 1.48 (0.75) | 0–4 |
| 2) Low cognitive functioning | 0.501*** | *0.78* | | | | | 1.19 (0.66) | 0–4 |
| 3) Age (in years) | -0.322*** | -0.103*** | —ᵃ | | | | 53.41 (15.63) | 18–89 |
| 4) Religiosity | -0.042 | 0.016 | 0.105*** | —ᵃ | | | 3.51 (3.32) | 0–10 |
| 5) Secondary education (Ref. Lower) | -0.032 | -0.129*** | -0.150*** | -0.016 | —ᵃ | | 0.56 (0.50) | 0–1 |
| 6) Female (Ref. male) | 0.185*** | 0.137*** | -0.197*** | 0.063* | -0.069* | —ᵃ | 0.44 (0.50) | 0–1 |
| **Experiment 2 (*N* = 1,089)** | | | | | | | | |
| 1) Stress | *0.77* | | | | | | 1.48 (0.75) | 0–4 |
| 2) Low cognitive functioning | 0.491*** | *0.78* | | | | | 1.18 (0.66) | 0–4 |
| 3) Age (in years) | -0.269*** | -0.013 | —ᵃ | | | | 54.23 (14.75) | 18–89 |
| 4) Religiosity | -0.031 | 0.062* | 0.110*** | —ᵃ | | | 3.63 (3.24) | 0–10 |
| 5) Secondary education (Ref. Lower) | -0.047 | -0.184*** | -0.149*** | -0.040 | —ᵃ | | 0.54 (0.50) | 0–1 |
| 6) Female (Ref. male) | 0.176*** | 0.080** | -0.162*** | 0.075* | -0.103*** | —ᵃ | 0.46 (0.50) | 0–1 |

Notes:

*$p<0.05$,

**$p<0.01$,

***$p<0.001$.

ᵃ Cronbach's alpha not applicable.

converted to vouchers, a ticket for a charity lottery, or donated to UNICEF. Written informed consent was obtained from all study participants.

In Experiment 1 on BSDs ($N_{Experiment\ 1}$ = 1,090 valid responses and usable data), the mean age of participants was 53 ($SD$ = 15.62), with 44.4% being female and 55.5% holding a high school diploma (Table 1 provides descriptive statistics of all independent variables of both experiments). In Experiment 2 on BCIs ($N_{Experiment\ 2}$ = 1,089 valid), the mean age of participants was 54 ($SD$ = 14.75), with 46.5% being female and 53.9% holding a high school diploma. Both experiments have been approved by the ethics committee of the University of Erfurt (reference number: EV-20190917). Although our study is not a medical study, we adhere to the Code of Ethics of the World Medical Association (Declaration of Helsinki) to protect human research participants [51].

## Experimental factorial survey with vignettes

We employed factorial survey designs with vignettes in which descriptions of hypothetical situations were experimentally varied. This is useful when "real world" manipulations are ethically or practically challenging [52, 53]. Such factorial surveys combine the advantages of experiments (e.g., high internal validity) with those of survey research (e.g., high external validity due to more reliable samples) [54–56]. For both experiments, between-subject designs were used in which each respondent was randomly assigned to one vignette that describes an NTD, its potential areas of application, and a graphic that explains the basic functions [cf., 8, 17]. Each 2x2x2 experiment varied the purpose of use (*treatment* vs. *enhancement*), the invasiveness of the NTD (*noninvasive* vs. *invasive*), and the question order (*moral acceptability before use willingness vs. the opposite order*) (S1 and S2 Figs in S1 File, as well as S3 and S4 Figs in S1 File for original German wording).

### Dependent variables

**Moral acceptability.** In the treatment purpose condition, respondents were asked about the moral acceptability of using the technology for medical reasons (e.g., prevention, diagnosis, or treatment of illnesses). In the enhancement purpose condition, they were asked about non-medical performance enhancement (e.g., using them in one's spare time or profession) (S1 and S2 Figs in S1 File). Response options ranged from "*completely unacceptable*" (0) to "*completely acceptable*" (4). Respondents could also choose the response option "*no response*" (selected by $N_{Experiment\ 1}$ = 21, or 1.85%, and $N_{Experiment\ 2}$ = 28, or 2.46%).

**Use willingness.** Similarly, respondents were asked about the willingness to use the technology for treatment or enhancement (S1 and S2 Figs in S1 File). Response options ranged from "*certainly not*" (0) to "*certainly*" (4). Again, respondents could choose "*no response*" (selected by $N_{Experiment\ 1}$ = 20, or 1.76%, and $N_{Experiment\ 2}$ = 21, or 1.85%).

### Independent variables

*Stress* was measured using three items of the German version [57] of the Perceived Stress Scale [58], a valid and reliable instrument to assess chronic stress. The items (e.g., "In the last 12 months, how often have you felt that you were unable to control important things in life?") had response options ranging from "*never*" (0) to "*very often*" (4). Internal consistency was satisfactory in both experiments (Table 1).

*Low cognitive functioning* was assessed with four items of the "Self-Organization and Problem Solving" subscale of the Deficits in Executive Functioning Scale [59]. An exemplary item is "Unable to 'think on my feet' or respond as effectively as others to unexpected events." Response options ranged from "*never*" (0) to "*very often*" (4). Measures revealed satisfactory internal consistency in both experiments (Table 1).

*Religiosity* was measured with the question "How important is religion to you?" on a scale ranging from "*not important*" (0) to "*very important*" (10) [cf., 60].

### Pretesting

By conducting think-aloud cognitive pretests (*N* = 12) with additional probing questions and a quantitative pretest (*N* = 34), we assessed and improved the comprehensibility and validity of all instructions, vignettes, and further instruments. This is particularly important given the complexity of NTDs and the possible difficulty in understanding the given information.

### Analysis

We used ordinary least squares regression models to explore how the experimental conditions (purpose of use, invasiveness, and question order) and respondent characteristics (stress, low cognitive functioning, religiosity, age, education, and gender) were associated with respondents' moral evaluation of and willingness to use BSDs and BCIs. After examining main effects, we tested for conditional effects of the use purpose by testing interaction effects between use purpose and each of the variables. All analyses were conducted with the statistical software package Stata 14.

## Results

### Descriptive results

Fig 1 shows that, regardless of the experimental condition, 25.5% and 28.7% of the participants considered BSDs and BCIs morally acceptable, respectively. While 14.6% of the respondents indicated that they would use BSDs with certainty, this share was slightly higher for BCIs

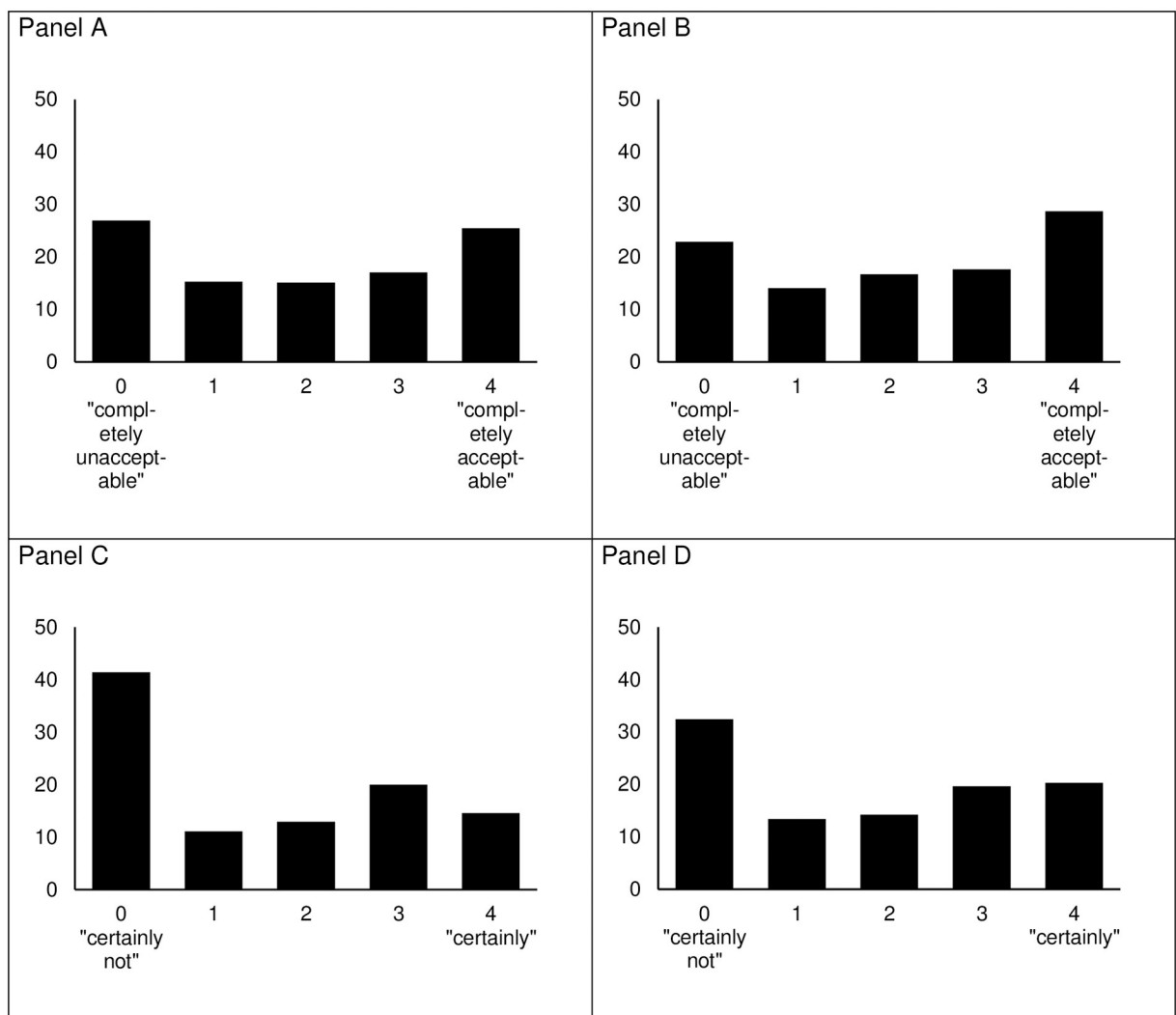

**Fig 1. Distribution of answers (in %) of the moral acceptability ($M$ = 1.99; $SD$ = 1.56, Panel A) and use willingness of BSDs ($M$ = 1.55; $SD$ = 1.53, Panel C) in Experiment 1 ($N$ = 1,090), and the moral acceptability ($M$ = 2.15; $SD$ = 1.54, Panel B) and use willingness of BCIs ($M$ = 1.82; $SD$ = 1.55, Panel D) in Experiment 2 ($N$ = 1,089).**

(20.3%). Moral acceptability and use willingness positively correlated for BSDs ($r$ = 0.81; $p$<0.001) and BCIs ($r$ = 0.83; $p$<0.001).

## Main effects analysis

Table 2 shows that using both types of NTDs for enhancement resulted in statistically significantly lower moral acceptability ($p_{BSD}$<0.001; $p_{BCI}$<0.001) and willingness to use ($p_{BSD}$<0.001; $p_{BCI}$<0.001) than using them for treatment. If the NTD was invasive (as compared to noninvasive), moral acceptability ($p_{BSD}$ = 0.002; $p_{BCI}$<0.001) and use willingness ($p_{BSD}$<0.001; $p_{BCI}$<0.001) were lower. Interestingly, if the moral acceptability was assessed prior to the use willingness, the moral acceptability of BCI ($p$<0.015) and the use willingness of both NTDs ($p_{BSD}$ = 0.002; $p_{BCI}$<0.002) were higher than when they were assessed in the opposite order. Moreover, education, stress, and low cognitive functioning were unrelated to the moral acceptability and use willingness of both NTDs (all $p$>0.05). While females did not

**Table 2. Ordinary least regression of the moral acceptability and use willingness regarding BSDs ($N_{Experiment\ 1}$ = 1,090) and regarding BCIs ($N_{Experiment\ 2}$ = 1,089).**

| | BSD: | BSD: | BCI: | BCI: |
|---|---|---|---|---|
| | Moral acceptability | Use willingness | Moral acceptability | Use willingness |
| Enhancement (Ref. Treatment) | -2.15*** | -2.22*** | -1.92*** | -2.09*** |
| | (0.07) | (0.06) | (0.07) | (0.07) |
| Invasive (Ref. Noninvasive) | -0.21** | -0.28*** | -0.35*** | -0.35*** |
| | (0.07) | (0.06) | (0.07) | (0.07) |
| Morality first (Ref. Second) | 0.08 | 0.19** | 0.17* | 0.21** |
| | (0.07) | (0.06) | (0.07) | (0.07) |
| Stress | 0.10 | 0.07 | -0.02 | 0.07 |
| | (0.06) | (0.05) | (0.06) | (0.05) |
| Low cognitive functioning | -0.05 | -0.05 | 0.06 | -0.03 |
| | (0.06) | (0.06) | (0.06) | (0.06) |
| Age | -0.00 | -0.01** | -0.01*** | -0.01*** |
| | (0.00) | (0.00) | (0.00) | (0.00) |
| Religiosity | -0.02* | -0.01 | -0.02* | -0.01 |
| | (0.01) | (0.01) | (0.01) | (0.01) |
| Secondary education (Ref. Lower) | 0.13 | 0.04 | 0.11 | 0.09 |
| | (0.07) | (0.06) | (0.07) | (0.07) |
| Female (Ref. Male) | -0.05 | -0.11 | -0.18* | -0.22** |
| | (0.07) | (0.07) | (0.07) | (0.07) |
| Constant | 3.30*** | 3.08*** | 3.88*** | 3.67*** |
| | (0.19) | (0.18) | (0.21) | (0.19) |
| *F-test* | 118.18*** | 145.57*** | 90.98*** | 120.00*** |
| Adjusted $R^2$ | 0.49 | 0.54 | 0.43 | 0.50 |

*Notes*: *B*-coefficients (standard errors in brackets);

*$p < 0.05$,

**$p < 0.01$,

***$p < 0.001$.

differ statistically in their moral acceptability ($p = 0.465$) and use willingness ($p = 0.080$) of BSDs, they were less accepting ($p = 0.013$) and less willing to use BCIs ($p = 0.002$) than males. Older respondents showed a lower moral acceptability of BCIs ($p < 0.001$) and a lower willingness to use both types of NTDs ($p_{BSD} < 0.003$; $p_{BCI} < 0.001$). Religiosity was negatively associated with the moral acceptability of both types of NTDs ($p_{BSD} = 0.021$; $p_{BCI} = 0.041$), but not willingness to use ($p_{BSD} = 0.128$; $p_{BCI} = 0.302$).

## Conditional effects of purpose of use

We explored whether the purpose effects were weaker or stronger conditional on the other experimental treatments analyzed and the respondents' characteristics. Here, we ran interaction effect models (S1-S4 Tables in S1 File). For conciseness, we report and visualize only the statistically significant interaction effects and report the full models in the Supplementary Information.

**Moral acceptability of BSDs.** We found a statistically significant negative interaction effect between purpose of use and invasiveness ($p = 0.001$, Model 1 in S1 Table in S1 File). Thus, invasive enhancements were less morally acceptable than noninvasive enhancements (Panel A, Fig 2). Panel B demonstrates a higher acceptability of enhancement by individuals

with higher stress than those with lower stress ($p$ = 0.001; Model 3). Those with lower cognitive functioning had a higher acceptance of enhancement than those with higher cognitive functioning. Treatment was less accepted if cognitive functioning was low ($p$ = 0.007, Model 4; Panel C). Enhancement was also less accepted with increasing age ($p$ = 0.002; Model 5; Panel D) and with higher levels of religiosity ($p$ = 0.020, Model 6; Panel E).

**Willingness to use BSDs.** The previously reported main effect of the question order of the willingness to use BSDs seemed to only occur for treatment. That is, when moral acceptability was assessed prior to use willingness, the willingness was higher than when the questions were presented in the opposite order ($p$ = 0.011, Model 2 in S2 Table in S1 File; Panel F). More stress ($p$ = 0.001, Model 3; Panel G) and lower cognitive functioning ($p$ = 0.003, Model 4; Panel H) increased the willingness to use NTDs for enhancement purposes, while the willingness to use it for treatment was slightly lower when more stress and higher cognitive functioning were reported.

**Moral acceptability of BCIs.** Similar to the BSD results, the acceptability of using BCIs for enhancement was higher for individuals who had higher stress than lower stress ($p$ = 0.041, Model 3 in S3 Table in S1 File; Panel I), but the acceptability for treatment was slightly lower when stress was high. The acceptability for enhancement purposes was lower with higher levels of religiosity ($p$ = 0.041, Model 6; Panel J). Females considered enhancement with BCIs less acceptable than males ($p$ = 0.036, Model 8; Panel K).

**Willingness to use BCIs.** Like BSDs, the acceptability of enhancement with BCIs increased with stress ($p$ = 0.021, Model 3 in S4 Table in S1 File; Panel L).

## Discussion

Various developments, such as digitalization, transformations in the work environment, and aging populations pose major challenges to societies [9, 61, 62]. The everlasting need to adapt to these changes can motivate individuals to use NTDs for enhancement (e.g., to meet work demands) and treatment (e.g., to enable impairment-free aging) purposes. To increase our understanding of the public evaluation of novel NTDs, we investigated in two experiments how the moral acceptability and willingness to use BSDs and BCIs were affected by different use purposes (treatment and enhancement), invasiveness (noninvasive and invasive methods), framing effects of the question order, and respondents' characteristics. For more in-depth insights, we tested if the purpose of use effects were conditioned by any of the included variables.

### Moderate acceptability and use willingness

Both experiments, on average, revealed a moderate moral acceptability and willingness to use BSDs and BCIs; about the same share of respondents completely morally accepted or rejected both NTDs. However, more respondents were definitely against using them than definitely for using them. Thus, the public neither fully embraces nor rejects NTDs [cf., 8, 12, 32]. Beyond this overall assessment, this study provides further insight into the specific conditions for attitudinal differences, thereby not overlooking more complex patterns of moral acceptance and willingness to use NTDs.

### Treatment preferred over enhancement

Our finding of a substantially higher moral acceptability and willingness to use BSDs and BCIs for treatment than enhancement [cf., 8, 25] suggests more enthusiasm towards NTDs for restoring "normal" functioning over achieving "superior" functioning. Respondents seemed to be more willing to accept potential risks and ethical concerns when using NTDs for a purpose

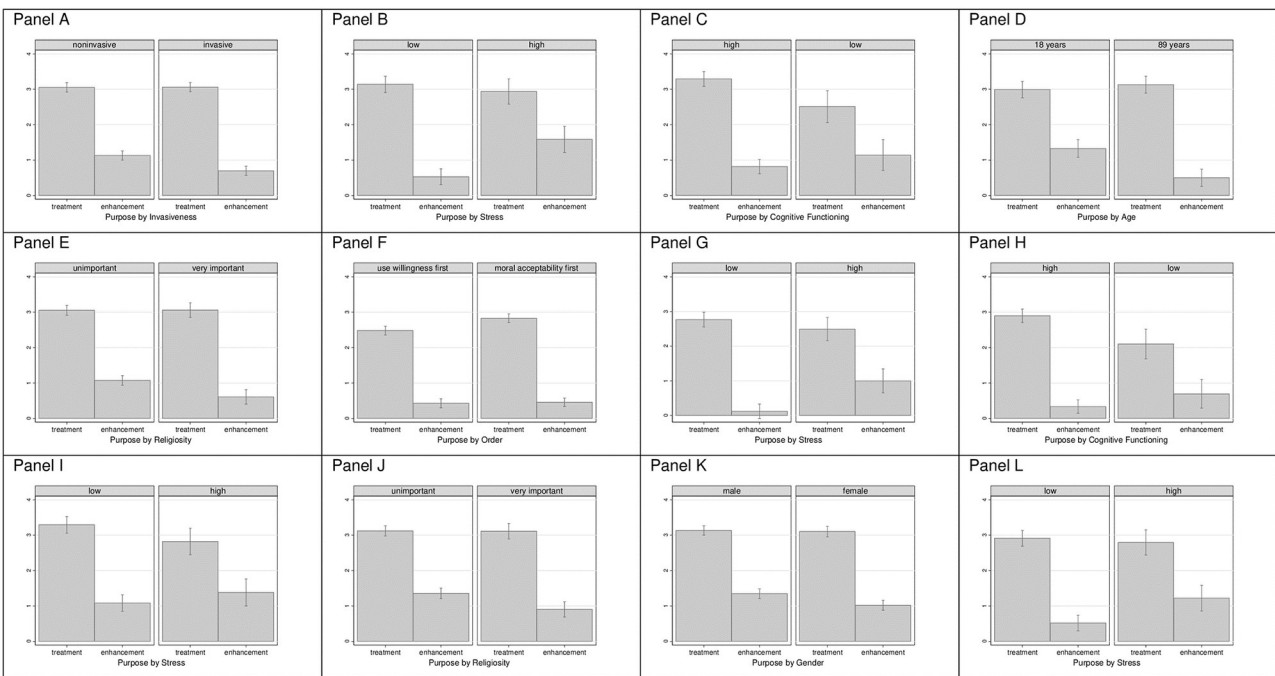

**Fig 2. Predicted values (with standard errors) for moral acceptability (Panels A to E) and use willingness (Panels F to H) of BSDs conditional on purpose of use and further factors in Experiment 1 (*N* = 1,090), and predicted moral acceptability (Panels I to K) and use willingness (Panel L) of BCIs conditional on purpose of use and further factors in Experiment 2 (*N* = 1,089).**

that was considered necessary to improve well-being than simply to enhance one's normal functioning [8, 33, 34]. But the constantly growing and often positive media portrayal of new NTDs might fuel enthusiasm for both purposes [63]. Additionally, the rapid development and sale of NTDs may indicate an increased supply and demand for them for nonmedical purposes [35, 36]. When prevalence data are absent, monitoring public attitudes, like their willingness, will offer insights about how prevalent this practice might become in the future and how urgent it will be to regulate NTDs that will be used for enhancement. It is important to understand how realistic public expectations are about using NTDs for treatment and enhancement, and whether policies should proactively highlight their limitations and dangers [63]. While an overall public enthusiasm seems unlikely, education and clear communication about the actual benefits and risks is needed for NTDs to be used safely (Delhove, Osenk, Prichard, & Donnelley, 2020).

## Hesitancy towards invasive devices

Our findings demonstrate that respondents were concerned about how NTDs are applied [10], since invasive BSD and BCI devices were evaluated more morally negatively than noninvasive and resulted in a lower willingness to use. Although invasiveness may decrease visibility and potential stigma, the hesitancy towards invasive devices may reflect the potential higher risks they carry or the moral and ethical concerns involved, such as implications for personhood or unnaturalness [5, 40, 64]. While the use of invasive devices was not judged differently for treatment, the moral acceptability was lower when using BSDs for enhancement [25, 38]. The risk of using NTDs as treatment could potentially be accepted as a means to an end, but respondents were more cautious when it was used for enhancement.

## Correlation between both attitudes and question order effect

The substantial correlation between the moral acceptability and willingness to use both devices [25, 29, cf., 65] could indicate that moral concerns are antecedents of intended or actual behavior. It can be argued that acting against moral concerns can evoke psychological consequences; acting in line with them can lead to positive emotions [66–68]. Moreover, moral evaluations can reflect how individuals define situations, which often consciously and unconsciously guide decision-making [69]. This is interesting, since asking respondents first about their moral evaluations yielded greater willingness to uses both NTDs and greater moral acceptability for BCIs. The moral assessment of BSDs did not change depending on the question order, pointing to a more stable morality for this NTD, while the willingness to use BSDs for treatment (but not for enhancement) was even higher if asking about morality first. Thus, first thinking about the moral acceptability of a behavior does not necessarily activate concerns that reduce the behavioral willingness, as previously found [41]. If respondents were informed about possible applications of NTDs for treatment (as in our study), they may appraise the technology more positively and express fewer concerns. This needs to be explored further.

## Interindividual differences in attitudes towards NTDs

One general observation is that our indicators for a potential need of more respondents for NTDs had no main effects. But the fact that they modulated the assessment of NTDs regarding different use purposes underlines the importance of our more detailed analyses. Respondents who had greater stress gave moral acceptance for both types of NTDs more readily and were more willing to use BSDs and BCIs for enhancement purposes. While this relationship has thus far been investigated only for prescription drugs [29, 42–44], BSDs and BCIs might also be seen as instruments to cope with stress and its negative consequences, as well as increase performance and manage stressful demands.

We also observed this pattern at lower levels of cognitive functioning, in which NTDs might be perceived as a means to catch up [70] and normalize performance [2]. Indeed, people with lower abilities seem to benefit more from NTDs than people with a higher baseline performance [71, 72]. Surprisingly, at higher levels of cognitive functioning, this pattern was reversed when NTDs were used for treatment purposes. One interpretation of this could be that there is a fear of losing higher functioning and therefore NTDs are accepted and used to retain it. Since more NTDs that may compensate for potential deficits are in development, the expectation and willingness to use them may grow. Moreover, similar to previous research [17, 41], younger individuals showed a higher acceptability (especially regarding BSDs for enhancement) and use willingness of NTDs. Older cohorts may have concerns due to generally slower or missing adoption of technological innovations like NTDs [73], although they could help with age-related decline in functioning. Thus, the results do not support the notions in discourse on the responsibility of using NTDs as instruments to counteract the suffering from age-related diseases [47]. However, younger individuals seem to view neurotechnological means to become better than "normal" as more justified, which may represent a greater openness to try novel technologies for certain purposes.

We observed a stronger disapproval of both NTDs for enhancement purposes among more religious individuals [4, 8, 17]. This might be because religiosity often aligns with "traditional" worldviews, where the observed effects could occur due to the belief that using NTDs in medically unnecessary situations is "unnatural" and transgresses boundaries created by God [17, 48]. Prior research also shows that concerns towards enhancement, and BCIs in particular, were stronger among religious individuals [17, 65]. Interestingly, the willingness to use NTDs

was uncorrelated to religiosity, which could indicate double standards. Although against one's values, NTDs might be used as an end to a mean.

Finally, our results provide further evidence for gender differences in accepting technology. Female respondents voiced less favorable attitudes towards BCIs, especially for enhancement purposes. This adds to findings suggesting that females are more concerned about technology [8, 17, 41]. Moreover, it may also match gender differences in self-confidence, anxiety, and motivation in using technology that have also been found to influence the acceptance of its use [74–76].

## Strengths, limitations, directions for future research

We used an experimental approach to test causal hypotheses regarding public attitudes towards two types of neurotechnologies. The multi-stage random selection process of adults increased generalizability. However, our experiments were not without limitations. Questions about moral acceptability and willingness to use NTDs, especially for enhancement purposes, are sensitive and can be subject to bias, but using vignettes has been shown to reduce issues of social desirability [77, 78]. We believe that the low share of "no response" in each variable (<2.5%) and low dropout rates (<0.4%) indicated limited problems with this. Additional analysis also showed that the outcome variables were not affected by anonymity perceptions towards the survey [79] and the effects of the other variables did not change when controlling for anonymity perceptions (S5 to S9 Tables in S1 File).

Still, the willingness of use measures can be seen as a limitation because of restrictions (such as money, time, skills, and opportunity) around voicing a willingness and actual behavior [28, 80]. However, willingness measures and behavior seem to substantially correlate [66, 81] and treatment effects in factorial surveys were comparable to other designs [82].

Although there might still be few users of NTDs, we encourage future research to examine differences in attitudes of those with personal or vicarious experience with them. Therefore, research on participant groups could offer insights about experienced or expected benefits and problems of NTDs, and the extent to which they may affect the lives of people around NTD users [3, 10]. Given the constantly changing field of NTDs and the potential increased awareness through the media [63], we suggest further observations of public attitudes to evaluate how individuals in- and outside of Germany assess these NTDs, and whether they are aware of and engage in beneficial or harmful uses. Thereby, future studies may compare international samples to examine if views about NTDs differ systematically and, if so, why. Such studies would help understand whether this study's findings using a representative sample of the adult population in Germany can be generalized to other contexts.

Finally, while this study relies on survey data collected online, we cannot exclude the possibility that our data are biased towards people who usually interact with basic technology. This can result, for example, in more positive views towards technologies. However, Internet access in Germany is widespread (95% percent of all households, according to Statista 2020 [83]), and our sampling strategy used a multi-stage random process based on an ADM telephone master. Still, to exclude or explicitly examine mode effects, future studies might use other recruitment strategies and survey modes (e.g., mail surveys).

## Conclusion

The number of NTDs and their range of medical and nonmedical applications is on the rise and will eventually affect more areas of life. Although these devices raise ethical challenges related to agency, identity, privacy, equality, normality, and justice [3, 15, 17, 64], systematic knowledge about evaluations of those using, potentially using, or rejecting NTDs is falling behind [10, 41]. Our results suggest that BSD and BCI devices are, for many respondents, not

the preferred means to achieve maximum functional enhancement, although the wish to enhance human functioning beyond "normal" levels is as old as humanity [9].

When medical necessity is lacking, respondents seem to have morally salient concerns about health and positional goods, as well as discomfort about NTDs creating excessive and beyond-normal functioning [33]. Our data therefore provide further evidence that public attitudes are influenced by a therapy-enhancement distinction, and these underlying concerns should be given due consideration in policy-making. Nevertheless, our results also show that a significant share of respondents was strongly accepting of both NTDs. This might be due to the thus far unobserved role of certain respondent characteristics, i.e., respondents who reported high stress levels and low cognitive functioning more readily accepted NTDs for enhancement. These results may signal their need for external helpers and more tolerance towards risky uses (which is difficult to observe when it occurs privately). This higher acceptance may deserve attention when deliberating about informing the public about NTDs and their regulation, since people might take more risks than wanted when dealing with external demands (e.g., from work). Older and more religious individuals were particularly skeptical regarding the use of NTDs, although older individuals especially could benefit from them. While treatment purposes were evaluated with more enthusiasm, especially by respondents with high cognitive functioning (who may wish to preserve their functioning with neurotechnological help), further investigation should examine how realistic those willing to use NTDs in a clinical setting evaluate their capabilities. Our findings that respondents are more reserved towards invasive than noninvasive NTDs serve as evidence that the potential risks of NTDs are not ignored. In sum, our study replicates several existing findings found in other populations or for other NTDs (e.g., stronger preferences for enhancement over treatment, or a preference for non-invasive over invasive NTDs), it also provides novel insights about public views on NTDs (e.g., moral framing effects for treatment purposes or interindividual differences). It is still essential to better understand the social and psychological mechanisms that drive societal norms, with a focus on the effective and ethically justifiable governance of NTDs. This knowledge can help better conceive and anticipate the public acceptance and future trends in the advancements of NTDs. It can help develop education about the nature and consequences of its use; reduce fears, misinformation, and misuse [20, 41]; and advance NTD development that is coherent with the values, priorities, and needs of the public [3, 10]. Our results aim to increase this understanding and suggest possible avenues for exploring the relationships between possible NTDs uses, ethical concerns, and public attitudes related to both.

## Supporting information

**S1 File.**
(DOCX)

## Acknowledgments

We thank those who helped to conduct this study, especially Guido Mehlkop (who also provided feedback on the manuscript), Floris van Veen, Fabian Hasselhorn, Saskia Huber, the forsa-team, and Kelsey Hernandez for language editing.

## Author Contributions

**Conceptualization:** Sebastian Sattler, Dana Pietralla.

**Data curation:** Sebastian Sattler, Dana Pietralla.

**Formal analysis:** Sebastian Sattler, Dana Pietralla.

**Funding acquisition:** Sebastian Sattler.

**Investigation:** Sebastian Sattler, Dana Pietralla.

**Methodology:** Sebastian Sattler, Dana Pietralla.

**Project administration:** Sebastian Sattler.

**Supervision:** Sebastian Sattler.

**Visualization:** Sebastian Sattler, Dana Pietralla.

**Writing – original draft:** Sebastian Sattler, Dana Pietralla.

**Writing – review & editing:** Sebastian Sattler, Dana Pietralla.

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
