## [Decision Letter · Decision Letter 0]

29 Jul 2022

PONE-D-22-11587Public attitudes towards neurotechnology: Findings from two experiments concerning Brain Stimulation Devices (BSDs) and Brain-Computer Interfaces (BCIs)PLOS ONE

Dear Sebastian,

Thank you for submitting your manuscript to PLOS ONE. After careful consideration, we feel that it has merit but does not fully meet PLOS ONE’s publication criteria as it currently stands. Therefore, we invite you to submit a revised version of the manuscript that addresses the points raised during the review process.

Please submit your revised manuscript by Sep 12 2022 11:59PM If you will need more time than this to complete your revisions, please reply to this message or contact the journal office at plosone@plos.org. Please include the following items when submitting your revised manuscript:A rebuttal letter that responds to each point raised by the academic editor and reviewer(s). You should upload this letter as a separate file labeled 'Response to Reviewers'.A marked-up copy of your manuscript that highlights changes made to the original version. You should upload this as a separate file labeled 'Revised Manuscript with Track Changes'.An unmarked version of your revised paper without tracked changes. You should upload this as a separate file labeled 'Manuscript'.

We look forward to receiving your revised manuscript.

Kind regards,

Mariella Pazzaglia

Academic Editor

PLOS ONE

Journal Requirements:

Additional Editor Comments:

Thank you for submitting your paper to PLOS ONE. I do have comments on your paper from two experts and have read it closely myself. I believe that there is value in these data, and that they should be documented in the literature. However, the reviewer has questioned on some concerns about replicability and study contribution and I share these reservations. I invite you to revise the paper in light of comments.

Reviewers' comments:

Reviewer's Responses to Questions

**Comments to the Author**

1. Is the manuscript technically sound, and do the data support the conclusions?

Reviewer #1: Partly

Reviewer #2: Yes

2. Has the statistical analysis been performed appropriately and rigorously? 

Reviewer #1: Yes

Reviewer #2: Yes

3. Have the authors made all data underlying the findings in their manuscript fully available?

Reviewer #1: Yes

Reviewer #2: Yes

4. Is the manuscript presented in an intelligible fashion and written in standard English?

Reviewer #1: Yes

Reviewer #2: Yes

5. Review Comments to the Author

Reviewer #1: This manuscript explores acceptability of two types of neurological devices, namely, brain stimulation devices and brain computer interfaces. The authors study their acceptance for both treatment and enhancement purposes and provide insights that might inform technology developers about people’s needs and concerns around ethical debates. The authors’ study particularly focusses on investigating whether the moral acceptability and the willingness to use those technologies depends on the purpose of use, their invasiveness, and framing effects.

This paper is overall well written and easy to read. I agree with the authors that we need a better understanding of people’s acceptability of new technologies, especially when there are ethical concerns around innovations. I appreciate the authors’ motivation on debate among scientists when it comes to medical vs enhancement uses. However, I have some concerns about replicability and study contribution that prevents me from being confident of recommending acceptance. I explain in detail below.

REPLICABILITY

-The authors motivate the need for further understanding on how the public evaluates different uses of NTDs, however their study is limited to a reduced population (a sample of population in Germany). How can the findings be extended or generalised to a broader community? Also, it is unclear what authors mean by “the public”, how broad this population should be?

-The study was a web-based survey which might be biased towards people who usually interact with basic technology. How could this affect the generality of the findings?

-I was missing more discussion on the effect of participants age on the findings. Participants age was quite diverse (18 - 89), so I wonder whether the participant age could influence their willingness to use novel technologies. Younger people might be more willing to try new things than older people. More discussion is needed.

STUDY CONTRIBUTION

I understand the need to further understand the role of framing effects, and I appreciate the authors’ contribution on replicating framing effects and testing whether such effects also exist for treatment purposes. Similarly, I appreciate the need for understanding interindividual differences in attitudes towards NTDs as there is unclarity in the literature, which stated by authors “research has yielded inconclusive results and has hardly investigated whether attributes of the NTDs are judged differently by different people [10,38]”

However, it seems to me that the role of the purpose and invasiveness was known already in the literature. For example, authors’ finding that “Respondents preferred treatment over enhancement purposes” seems to only replicate previous work, as stated by authors on lines 88-89: “While people more-readily accepted risks and ethical concerns when using NTDs for treatment [34], using them for enhancement seems to be perceived more negatively [8,25]”, and on lines 95-96: “Despite the growing interest in using NTDs for enhancement, we expect a higher acceptability and willingness to use them for treatment [9,25,32,33]”. This applies for the finding that “Respondents preferred noninvasive over invasive devices” as well. Therefore, it’s unclear the contribution of the present study in the context of purpose and invasiveness.

Reviewer #2: The article investigates public attitudes towards neurotechnologies for medical and non-medical conditions. Overall, I found the article interesting and conducted with rigorous methodology, though in some points difficult to follow. Here some recommendations to improve the readability:

1. Abstract: please provide a brief explanation of "framing effects", like "... depending on their invasiveness, framing effects (variable order of assessing moral acceptability and willingness to use), and interindividual differences..."

2. Abstract and main text: please use the same order of variables investigated, to be consistent and facilitate readers on following the flow of the article. For instance regarding variables investigated: (1): purpose; (2) invasiveness; (3) framing effects; (4) personal factors/independent variables associated..

3. Table 1: I'm sorry but I find the interpretation of the table difficult. The columns 1), 2), 3)... refers to the same variables in rows (1) Stress, 2) Low cognitive functioning, 3)...)? In this case, correlations between, for instance, "1) Stress" should not be a perfect correlation (1.00)? Please clarify.

6. PLOS authors have the option to publish the peer review history of their article (what does this mean?). If published, this will include your full peer review and any attached files.

Reviewer #1: No

Reviewer #2: No

---

## [Author Response · Author response to Decision Letter 0]

14 Aug 2022

Dear Reviewers,

Thank you for taking the time and effort to read our paper. We appreciate your positive feedback and constructive criticism. We believe that incorporating the comments enriched and strengthened the manuscript.

In the following pages, you will find our responses to your general and specific comments.

Reviewer #1:

This manuscript explores acceptability of two types of neurological devices, namely, brain stimulation devices and brain computer interfaces. The authors study their acceptance for both treatment and enhancement purposes and provide insights that might inform technology developers about people’s needs and concerns around ethical debates. The authors’ study particularly focusses on investigating whether the moral acceptability and the willingness to use those technologies depends on the purpose of use, their invasiveness, and framing effects.

This paper is overall well written and easy to read. I agree with the authors that we need a better understanding of people’s acceptability of new technologies, especially when there are ethical concerns around innovations. I appreciate the authors’ motivation on debate among scientists when it comes to medical vs enhancement uses. However, I have some concerns about replicability and study contribution that prevents me from being confident of recommending acceptance. I explain in detail below.

COMMENT 1:

The authors motivate the need for further understanding on how the public evaluates different uses of NTDs, however their study is limited to a reduced population (a sample of population in Germany). How can the findings be extended or generalised to a broader community? Also, it is unclear what authors mean by “the public”, how broad this population should be?

OUR RESPONSE TO COMMENT 1:

We would agree that findings beyond Germany would support generalizing the findings and that international studies are beneficial for that; however, we would also argue that single-country studies have merits as they allow to place particular attention on a certain context (i.e., the general population in Germany in the age of 18 and older). To acknowledge the point of the reviewer, we added the following statement to the Limitations section: “we suggest further observations of public attitudes to evaluate how individuals in- and outside Germany assess these NTDs, and whether they are aware of and engage in beneficial or harmful uses. Thereby, future studies may compare international samples to examine if views about NTDs differ systematically and, if so, why. Such studies would help understand whether this study’s findings using a representative sample of the adult population in Germany can be generalized to other contexts.”

The public investigated here is described in the Current Study section: “By investigating two NTDs, this study facilitates a better understanding of how public attitudes in Germany relate to both NTDs and the investigated factors. Using randomly selected samples of the general adult population in Germany…”

COMMENT 2:

The study was a web-based survey which might be biased towards people who usually interact with basic technology. How could this affect the generality of the findings?

OUR RESPONSE TO COMMENT 2:

We agree that web-based surveys could be biased towards people who usually interact with basic technology. However, internet access in Germany is very widespread (95% percent of all households, according to Statista 2020 [82]). Moreover, our sample was recruited offline. Still, we will include this as a limitation to our findings and propose a strategy for further studies to reduce this bias: “Finally, while this study relies on survey data collected online, we cannot exclude the possibility that our data are biased towards people who usually interact with basic technology. This can result, for example, in more positive views towards technologies. However, Internet access in Germany is widespread (95% percent of all households, according to Statista 2020 [82]), and our sampling strategy used a multi-stage random process based on an ADM telephone master. Still, to exclude or explicitly examine mode effects, future studies might use other recruitment strategies and survey modes (e.g., mail surveys).”

COMMENT 3:

I was missing more discussion on the effect of participants age on the findings. Participants age was quite diverse (18 - 89), so I wonder whether the participant age could influence their willingness to use novel technologies. Younger people might be more willing to try new things than older people. More discussion is needed.

OUR RESPONSE TO COMMENT 3:

We describe this finding in the results section: “Older respondents showed a lower moral acceptability of BCIs (p<0.001) and a lower willingness to use both types of NTDs (pBSD<0.003; pBCI<0.001).” Our findings also show that “Enhancement was also less accepted with increasing age (p=0.002; Model 5; Panel D).” These findings were explained in the Discussion section: “Moreover, similar to previous research [17,41], younger individuals showed a higher acceptability (especially regarding BSDs for enhancement) and use willingness of NTDs. Older cohorts may have concerns due to generally slower or missing adoption of technological innovations, like NTDs [72], although they could help with age-related decline in functioning. Thus, the results do not support the notions in discourse on the responsibility of using NTDs as instruments to counteract the suffering from age-related diseases [47]. However, younger individuals seem to view neurotechnological means to become better than “normal” as more justified, which may represent a greater openness to try novel technologies for certain purposes.” So, indeed, younger people were found to be more willing to use NTDs as you suggest.

COMMENT 4:

I understand the need to further understand the role of framing effects, and I appreciate the authors’ contribution on replicating framing effects and testing whether such effects also exist for treatment purposes. Similarly, I appreciate the need for understanding interindividual differences in attitudes towards NTDs as there is unclarity in the literature, which stated by authors “research has yielded inconclusive results and has hardly investigated whether attributes of the NTDs are judged differently by different people [10,38]”

However, it seems to me that the role of the purpose and invasiveness was known already in the literature. For example, authors’ finding that “Respondents preferred treatment over enhancement purposes” seems to only replicate previous work, as stated by authors on lines 88-89: “While people more-readily accepted risks and ethical concerns when using NTDs for treatment [34], using them for enhancement seems to be perceived more negatively [8,25]”, and on lines 95-96: “Despite the growing interest in using NTDs for enhancement, we expect a higher acceptability and willingness to use them for treatment [9,25,32,33]”. This applies for the finding that “Respondents preferred noninvasive over invasive devices” as well. Therefore, it’s unclear the contribution of the present study in the context of purpose and invasiveness.

OUR RESPONSE TO COMMENT 4:

We agree that our study – next to novel findings – replicates certain effects that have been found for other neurotechnologies (such as prescription drugs) or in other populations (e.g., non-representative samples, such as M-Turk participants or students). We are convinced that replications of findings are still undervalued in scientific practices; they deserve more attention to avoid singular findings. Moreover, our study provides several novel findings or rarely studied effects, several of which have been already named by the reviewer (e.g., testing framing effects for treatment purposes or examining interindividual differences in attitudes towards NTDs). Even alone, we believe these findings are important to publish because interindividual differences in treatment effects within experimental designs have hardly been tested, and because several prior studies relied on non-representative samples, while ours uses a representative sample. Moreover, by examining brain-computer interfaces in addition to brain stimulation, we examine effects for a neurotechnology that has received less attention in the past and for which the mentioned effects have not been researched (to our knowledge). Additionally, we tested factors that, to our knowledge, have not been tested before (such as low cognitive functioning of respondents). We are, therefore, convinced that our study contributes to the body of knowledge. To be clearer to readers, we added the following sentence in the Conclusion section: “In sum, our study replicates several existing findings found in other populations or for other NTDs (e.g., stronger preferences for enhancement over treatment, or a preference for non-invasive over invasive NTDs); it also provides novel insights about public views on NTDs (e.g., moral framing effects for treatment purposes or interindividual differences).”

Reviewer #2:

The article investigates public attitudes towards neurotechnologies for medical and non-medical conditions. Overall, I found the article interesting and conducted with rigorous methodology, though in some points difficult to follow. Here some recommendations to improve the readability:

COMMENT 1:

Abstract: please provide a brief explanation of "framing effects", like "... depending on their invasiveness, framing effects (variable order of assessing moral acceptability and willingness to use), and interindividual differences..."

OUR RESPONSE TO COMMENT 1:

We followed your recommendation and added a brief explanation: “Using vignettes describing how two NTDs, brain stimulation devices (BSDs; NExperiment 1=1,090) and brain-computer interfaces (BCIs; NExperiment 2=1,089), function, we randomly varied the purpose (treatment vs. enhancement) and invasiveness (noninvasive vs. invasive) of the NTD, and assessed framing effects (variable order of assessing moral acceptability first vs. willingness to use first).”

COMMENT 2:

Abstract and main text: please use the same order of variables investigated, to be consistent and facilitate readers on following the flow of the article. For instance regarding variables investigated: (1): purpose; (2) invasiveness; (3) framing effects; (4) personal factors/independent variables associated..

OUR RESPONSE TO COMMENT 2:

We checked for the consistency of the order.

COMMENT 3:

Table 1: I'm sorry but I find the interpretation of the table difficult. The columns 1), 2), 3)... refers to the same variables in rows (1) Stress, 2) Low cognitive functioning, 3)...)? In this case, correlations between, for instance, "1) Stress" should not be a perfect correlation (1.00)? Please clarify.

OUR RESPONSE TO COMMENT 3:

The first row of the table clarifies this: “Pairwise correlations and Cronbach’s α in the diagonal”. We have now put the specific information about the diagonal in italics (as well as the numbers in the table). Moreover, the table now notes more clearly the meaning of the diagonal: “a Cronbach’s α not applicable.”

---

## [Editor Report · Decision Letter 1]

19 Sep 2022

Public attitudes towards neurotechnology: Findings from two experiments concerning Brain Stimulation Devices (BSDs) and Brain-Computer Interfaces (BCIs)

PONE-D-22-11587R1

Dear Dr. Sattler,

We’re pleased to inform you that your manuscript has been judged scientifically suitable for publication and will be formally accepted for publication once it meets all outstanding technical requirements.

Kind regards,

Mariella Pazzaglia

Academic Editor

PLOS ONE